# Automatic Liver Tumor Segmentation from CT Images Using Graph Convolutional Network

**DOI:** 10.3390/s23177561

**Published:** 2023-09-01

**Authors:** Maryam Khoshkhabar, Saeed Meshgini, Reza Afrouzian, Sebelan Danishvar

**Affiliations:** 1Faculty of Electrical and Computer Engineering, University of Tabriz, Tabriz 51666-16471, Iran; 2Miyaneh Faculty of Engineering, University of Tabriz, Miyaneh 51666-16471, Iran; 3College of Engineering, Design, and Physical Sciences, Brunel University London, Uxbridge UB8 3PH, UK

**Keywords:** Chebyshev graph convolution, CT images, deep learning, liver segmentation

## Abstract

Segmenting the liver and liver tumors in computed tomography (CT) images is an important step toward quantifiable biomarkers for a computer-aided decision-making system and precise medical diagnosis. Radiologists and specialized physicians use CT images to diagnose and classify liver organs and tumors. Because these organs have similar characteristics in form, texture, and light intensity values, other internal organs such as the heart, spleen, stomach, and kidneys confuse visual recognition of the liver and tumor division. Furthermore, visual identification of liver tumors is time-consuming, complicated, and error-prone, and incorrect diagnosis and segmentation can hurt the patient’s life. Many automatic and semi-automatic methods based on machine learning algorithms have recently been suggested for liver organ recognition and tumor segmentation. However, there are still difficulties due to poor recognition precision and speed and a lack of dependability. This paper presents a novel deep learning-based technique for segmenting liver tumors and identifying liver organs in computed tomography maps. Based on the LiTS17 database, the suggested technique comprises four Chebyshev graph convolution layers and a fully connected layer that can accurately segment the liver and liver tumors. Thus, the accuracy, Dice coefficient, mean IoU, sensitivity, precision, and recall obtained based on the proposed method according to the LiTS17 dataset are around 99.1%, 91.1%, 90.8%, 99.4%, 99.4%, and 91.2%, respectively. In addition, the effectiveness of the proposed method was evaluated in a noisy environment, and the proposed network could withstand a wide range of environmental signal-to-noise ratios (SNRs). Thus, at SNR = −4 dB, the accuracy of the proposed method for liver organ segmentation remained around 90%. The proposed model has obtained satisfactory and favorable results compared to previous research. According to the positive results, the proposed model is expected to be used to assist radiologists and specialist doctors in the near future.

## 1. Introduction

The liver, which aids in food digestion, is the primary organ located below the right ribs and behind the base of the lung [1]. It controls storage, nutritional recovery, and blood cell filtration [2]. The right and left lobes of the liver are divided into two main regions. Two other lobe types are the caudate and quadrate. The hepatocellular carcinoma etiology is comparable to the liver cells’ fast growth and potential for spreading to other body parts [3]. Primary hepatic cancers develop when cells behave erratically [4]. Infection frequency among men is around double that of women worldwide [5].

According to World Health Organization (WHO) statistics, cancer claimed the lives of 8.8 million individuals in 2015, with 788,000 dying from liver cancer. The American Cancer Society (ACS) recently identified approximately 40,710 new cancer cases (29,200 men and 11,510 women), of which 28,920 people (19,610 men and 9310 women) were diagnosed with malignant liver and bile duct cancer and will die soon [6]. Liver cancer has recently become more common in Sub-Saharan Africa and East Asia, with over 600,000 people dying in 2017. Liver tumors are one of the most serious diseases that can endanger a person’s life and health. Male mortality from liver cancer is the second most prevalent, and female mortality from liver cancer is the sixth most common.

Sonography with ultrasound waves, computed tomography (CT), positron emission tomography (PET), magnetic resonance imaging (MRI), and invasive sampling are used to identify and categorize liver abnormalities and tumors. Hemangioma, localized nodular hyperplasia, adenoma, hepatocellular carcinoma, intrahepatic cholangiocarcinoma, and liver metastases are all described using these techniques [7]. Each of the techniques listed has benefits and drawbacks. However, because of the trade-off in terms of speed and precision between various techniques, CT imaging is suggested for the identification and segmentation of liver tumors. In addition, CT scans are used in medical image segmentation, such as vessel contour detection (VCD) in intravascular images and three-dimensional (3D) cross-modality cardiac image segmentation for cardiac disease diagnosis and treatment [8,9]. CT imaging is accessible in most medical facilities, and three-dimensional pictures of liver parts can be obtained quickly [10]. However, detecting the liver and separating tumors from CT pictures is a difficult job that necessitates the expertise of radiologists and specialized physicians. It is also time-demanding and mistake-prone. This is due to the following reasons: First, the form and texture of the liver vary across the CT image. Second, the contrast in CT images for nearby tissues and organs is minimal. Third, different artifacts in the acquired CT images may cause the liver organ to appear vague, indistinct, and non-uniform. Furthermore, a tissue may be misdiagnosed as a tumor, or a tumor may not be identified, putting the patient’s life in peril. As a result, greater care and focus are required when segmenting hepatic lesions in CT scans [11]. Consequently, developing an intelligent technique for autonomously segmenting liver lesions appears essential. As a result, many methods based on machine learning and image processing techniques have been created in recent years to identify the liver and liver lesions and have given satisfactory results; however, these methods need to be improved before entering the actual field and the real world [12]. In the following, some recent studies are reviewed to identify the liver organ and segmentation of liver tumors, along with examining the advantages and disadvantages. In the next paragraph, research based on MRI images is reviewed; then, studies based on CT will be reviewed in the following paragraph.

Zheng et al. [13] used MRI images to automate liver tumor segmentation. These experts showed a model that combined Long Short-Term Memory (LSTM) networks with CNN. Their suggested model makes use of 4D information found in MRI scans. Furthermore, the researchers’ suggested model’s initial weights were arranged using the U-Net network. Based on their research, the Dice similarity coefficient (DSC) and volumetric resemblance were 84% and 89%, respectively. Hänsch et al. [14] presented a new model using MRI images to segment liver cancer tumors. Based on this, these researchers used a heterogeneous 3D U-Net architecture. The DSC and F1 score reported by these researchers were 70% and 59%, respectively. The low accuracy of classification and low speed of segmenting cancer tumors were among the disadvantages of this research.

Ahmad et al. [15] presented a new method for automatically segmenting liver tumors based on deep learning networks. For this purpose, these researchers used a Convolutional Neural Network (CNN) with three CNN layers and two fully connected layers. One of the benefits of this study was the fast image segmentation. Rahman et al. [16] proved a novel algorithm for automatically segmenting liver tumors from CT images. These researchers’ model was built on a combination of ResNet and U-Net networks. They evaluated their suggested model, ResUNet, on the IRCADB01 3D collection and found it to be 97% accurate. Manjunath et al. [17] introduced a novel deep model for automated liver tumor segmentation based on CT images. These researchers’ suggested model is based on a modified model of the customized U-Net network that was improved with ResNet. According to the LiTS17 dataset, the DSC for tumor segmentation and liver diagnosis is 96% and 89%, respectively. This assessment measure was also 91% for liver diagnosis and 69% for tumor segmentation on the 3D IRCADb dataset. Di et al. [18] described a novel technique for automatically segmenting liver tumors. As a result, these researchers incorporated CT scans into their suggested network. For liver tissue extraction, they suggested a 3D U-Net network. The images are split into homogeneous superpixels after a hierarchical segmentation approach extracts the liver area, and these superpixels are iteratively decomposed based on their standard deviation intensities. Finally, a Support Vector Machine (SVM) splits each pixel into tumor and non-tumor groups based on local intensity and tissue traits. 

Dickson et al. [19] presented a new method for automatically segmenting liver tumors from CT images. In this paper, these researchers used a deep learning based on the improved U-Net network for tumor segmentation. Tummala et al. [20] described a novel algorithm for detecting liver tumors in CT images. The model developed by the researchers was built on an encoder/decoder architecture that could divide liver tumors into two steps. In this study, the DSC was found to be 65%. Sabir et al. [21] presented a new model based on deep learning networks for target tissue detection and tumor segmentation. The model proposed by these researchers was evaluated on the IRCADb01 3D dataset and had a DSC of 97% for liver organ detection and 83% for liver tumor segmentation. Dong et al. [22] presented a new model based on deep learning networks for automatically segmenting liver tumors based on CT images. Accordingly, their proposed deep model was based on hybrid convolutional networks based on the pre-trained Inception network. The DSC for liver organ diagnosis was 97%, and for segmentation of liver tumors was reported as 87%. Tang et al. [23] used a two-stage deep network called E^2^NET to automatically detect liver and liver tumors. These researchers used the LiTS17 database to evaluate their proposed method. The model proposed by these researchers had a DSC of 78% for tumor segmentation and could perform better than hybrid models. Li et al. [24] used the CT scan images of the LiTS17database to automatically detect liver and liver tumors. They designed their proposed network using a variable hybrid structure based on U-Net. This research showed that the architecture proposed in this work was very efficient for detecting small-sized tumors. Ansari et al. [25] used a new deep network based on U-Net called RESNET to detect liver tumors automatically. In this study, the researchers used a database based on CT scan images. The IoU based on the proposed method of these researchers was around 92%. Khan et al. [26] used an automatic unregistered deep learning-based model with residual block and unfolded convolution to segment liver tumors. The results obtained based on their proposed method based on two databases of Dircadb and LiTS17 show DSC of 91% and 88%, respectively. Bogui et al. [27] presented a new method for segmenting liver tumors using CT scan images based on deep learning networks. The researchers improved the architecture of the U-Net network and designed a new network called U-NeXt. The results for the sensitivity based on their method are around 95%.

As is clear from the review of previous studies, despite the favorable efficiency, these studies have several basic limitations. The first limitation is using classical machine learning algorithms for feature selection/extraction and segmentation. Using these algorithms causes the selected feature vector to be not optimal for segmentation and classification. The next limitation of previous research can be considered low detection accuracy. According to previous research, the detection accuracy for the liver organ is nearly equal to the detection accuracy for humans. However, previous research has found that the accuracy of liver tumor diagnosis is low, has limitations, and needs to be improved. The next limitation of previous studies is that the algorithm used in these studies was evaluated under stable and noise-free environments, and the efficiency of these algorithms in noisy environments is unknown. The current study aims to overcome previous studies’ relevant challenges and limitations. In the proposed method, the Simple Linear Iterative Clustering (SLIC) algorithm is used to cluster CT images of the liver, and graph convolutional networks are used to detect liver organs and segment liver tumors. According to the obtained results, the proposed method can diagnose and segment liver and liver tumors with the precondition of high accuracy. The contribution of this study is as follows:Using SLIC algorithm for optimal clustering of CT images.Using deep graph convolutional networks to segment the liver and liver tumors for the first time.Evaluation of the proposed network in noisy environments and maintaining the algorithm’s stability for various SNRs.Providing customized architecture based on the combination of SLIC and Chebyshev graph convolutional network.Compared to previous research, achieving the highest accuracy in segmenting and diagnosing liver and liver tumors on the LiTS17 dataset.

The remainder of the paper is structured as follows: Section 2 examines the database used in this research. Also, this section examines the mathematical background related to the proposed algorithm. Section 3 examines the proposed method of this study along with relevant details. Section 4 presents the simulation results according to the proposed method and examines the proposed model with previous studies. Finally, Section 5 presents the conclusion.

## 2. Materials and Methods

This section comprehensively explains the LiTS17 database employed in this investigation. Also, this section will fully describe the mathematical background of graph convolutional networks. Metrics for evaluating segmentation and SLIC are also covered.

### 2.1. Database Settings

The LiTS17 (Liver Tumor Segmentation 2017) database is used in this research [28]. This database is a collection of 130 patients with the maximum number of CT slices equal to 623 for each patient. CT volumes of 10 patients are used in this research. Each volume contains a different number of slices. Slices containing liver tissue are considered for liver segmentation. In this investigation, nine volumes of the LiTS17 database were employed. After reading the .nii data, 4158 images and their associated mask were accessible. The masks corresponding to the liver organ should be extracted from the 4158 images, and lastly, 987 images were utilized to train the network. The dimensions of each image are considered equal to 512 × 512. An example of a CT image with a corresponding mask of the tumor and liver is shown in Figure 1.

### 2.2. Graph Convolution

This part provides an introduction to graph convolution. The research direction of Michaël Defferrard et al. [29] led to the popularization of Graph Signal Processing (GSP). The GSP functions consider the graph’s structure and its parts’ attributes. In this study area, signal-processing techniques, including the Fourier transform, have been applied to the graphs to extend convolutions to the graph domain. Graph spectral filtering, commonly known as graph convolution, is developed using the Fourier transform in GSP [30].

The concept of graph convolution has been clarified. Correspondingly, let $D\in R^{N\times N}$ and $W\in R^{N\times N}$ stand for the graph’s diagonal degree and adjacency matrices. The degree matrix’s *i*-th diagonal component may be determined by
(1)$$D_{ii}=\sum_{i}W_{ij}$$

Subsequently, L, the graph’s Laplacian matrix, is represented as
(2)$$L=D-W\in R^{N\times N}$$

The eigenvectors of the graph’s Laplacian matrix are employed to compute the basic functions in the graph domain. The Singular Value Decomposition (SVD) could be used to obtain the eigenvectors of the graph Laplacian matrix U:(3)$$L=U\Lambda U^{T}$$

The columns $U=[u_{0},\ldots ,u_{N-1}]\in R^{N\times N}$ constitute the Fourier basis, and $\Lambda =diag([\lambda_{0},\ldots ,\lambda_{N-1}])$ is a diagonal matrix. The Fourier basis for the graph may be obtained by computing the eigenvectors of the Laplacian. The following is the (GFT) expression for a specific signal $x\in \;R^{N}$ representing the stacked feature vectors on the graph nodes:(4)$$U=[u_{0},\ldots ,u_{N-1}]\in R^{N\times N}$$
where $\hat{x}$ is the graph Fourier transforms’ solution because it is the converted signal in the frequency domain; Equation (4) it is possible to describe the inverse of GFT using the following form:(5)$$\hat{x}=U^{T}x$$
where $\hat{x}$ denotes the transformed signal in the frequency domain and is the Fourier transform graph’s solution; Equation (5) argues that the following may be used to express the inverse of GFT:(6)$$x=UU^{T}x=U\hat{x}$$

Using the outlined GFT, we can conduct the graph convolution operation and find the convolution of two signals on the graph domain. Convolution of the x and y on the graph *g:(7)$${x^{*}}_{g}y=U((U^{T}\hat{x})⊙(U^{T}y))$$
where ⊙, the element-wise Hadamard product, seems denoted by the symbol and is computed between the graph’s Fourier converted signals. Since a filtering function has been established, we can depict the filtered form of by (L) as
(8)y=g(L)x

The statement that follows in Equation (8) could potentially be used to illustrate how the signal graph convolution process and the filtering procedure in Equation (7) are related.
(9)$$\begin{array}{l} y=g(L)x=Ug(\Lambda )U^{T}x=U(g(\Lambda )).(U^{T}x)\\ =U(U^{T}(Ug(\Lambda ))).(U^{T}x)={x^{*}}_{g}(Ug(\Lambda )) \end{array}$$

### 2.3. Chebyshev Graph Convolution

We present the Chebyshev graph convolution, a specific form of graph convolution in which g(L), the Chebyshev polynomial expansion of L, replaces Equation (8). The graph convolution of *x* with is denoted by Ug(Λ): (10)$$y=g(L)x=g(U\Lambda U^{T})x=Ug(\Lambda )U^{T}x$$
where Equation (2) allows for the computation of W to provide the L, and Equation (3) allows for the calculation of Λ. With K-order Chebyshev polynomials, the g(Λ) is approximated. The normalized form allows one to approximate the g(Λ) function. Accordingly, $\Lambda_{\max}$ defines the greatest component among the diagonal entries of Λ, and the normalized value of Λ is as outlined below:(11)$$\tilde{\Lambda }=\frac{2\Lambda }{\Lambda_{\max}}-I_{N}$$
where the diagonal components of $\tilde{\Lambda }$ are in the range of [-1, 1] and $I_{N}$ is the N×N identity matrix. According to the framework of *K*-order Chebyshev polynomials, the following approximates g(Λ): (12)$$g(\Lambda )=\sum_{k=0}^{K-1}\theta_{k}T_{k}(\tilde{\Lambda })$$
where $\theta_{k}$ is the Chebyshev polynomial coefficient, and $T_{k}\left(\tilde{\Lambda }\right)$ may be retrieved by researching these phrases:(13)$$\left\{\begin{array}{l} T_{0}\left(\tilde{\Lambda }\right)=1,\;T_{1}\left(\tilde{\Lambda }\right)=\tilde{\Lambda }\\ T_{k}\left(\tilde{\Lambda }\right)=2(\tilde{\Lambda })(T_{k-1}(\tilde{\Lambda }))-T_{k-2}(\tilde{\Lambda })\;\;\;,\;\;\;k\geq 2 \end{array}\right.$$

It is demonstrated in Equation (14) that the graph convolution operation specified in Equation (12) may be represented by employing Equation (13):(14)$$\begin{array}{l} y=U\;g(\Lambda )\;\;U^{T}x\\ =\sum_{k=0}^{K-1}U\left[\begin{array}{l} \theta_{k}T_{k}({\tilde{\lambda }}_{0})\;\;\;\cdots \;\;\;\;\;\;\;\;\;\;0\\ \;\vdots \;\;\;\;\;\;\;\;\;\;\;\;\;\ddots \;\;\;\;\;\;\;\;\;\;\;\vdots \;\;\;\\ 0\;\;\;\;\;\;\;\;\;\;\;\;\cdots \;\;\;\;\theta_{k}T_{k}({\tilde{\lambda }}_{N-1}) \end{array}\right]\;U^{T}x\\ =\sum_{k=0}^{K-1}\theta_{k}T_{k}(\tilde{L})x \end{array}$$

The normalized Laplacian matrix is represented by $\tilde{L}=2\frac{L}{\lambda_{\max}}-I_{N}$. The Chebyshev graph convolution formula in Equation (14) [30] demonstrates that the two procedures are similar when combining the convolutional results for x with each Chebyshev polynomial component.

### 2.4. SLIC-Based Graph Embedding

The input image ***I*** was split into several uniformly disjoint sections ${\left\{R_{i}\right\}}_{i=1}^{N}$ by applying the SLIC superpixel technique to generate the region adjacency graph G (refer to Figure 2) [31]. N = P × Q/1000, the number of providing regions, was chosen randomly; P × Q stands for imagined spatial resolution. By dividing each channel by 2B − 1, where B indicates the bit depth of each channel, ref. [31] image properties were previously normalized to the range [0, 1]. The superpixels *Ri* are denoted by vertices vi in the resultant region adjacency network, each characterized by a one-dimensional feature vector *f_i_* obtained from the relevant area. The mean intensity for the region *R_i_* was taken into account by averaging the pixel intensities. Weighted edges linked the nodes that represented the adjacent regions. The following formula represents a Gaussian weighting function from which weights are derived: (15)$$w_{ij}=e^{-\beta d(f_{i-}f_{j})}$$
where
(16)$$d(f_{i}-f_{j})=\sum_{k=1}^{C}{\left(f_{i}^{k}-f_{j}^{k}\right)}^{2}$$
and β is a free parameter of the method.

A region adjacency matrix, A = [a*_ij_*] ∈ R*^N^*^×*N*^, conforms to the following:(17)$$a_{ij}=\left\{\frac{1\;\mathrm{if}\;\mathrm{regions}\;R_{i}\;\mathrm{and}\;R_{j}\;\mathrm{are}\;\mathrm{adjacent}}{0\;\;\;\;\;\;\;\;\;\;\;\;\;\;\;\;\;\;\;\;\;\;\;\;\;\;\;\;\;\;\;\;\;\;\;\;\;\;\;\;\;\;\;\;\;\;\;\;\;\;\;otherwise}\right.$$

As well, there is a region feature matrix, X ∈ R*^N^*^×1^, which represents the *N* vertices, each of which has features in R^1×1^.

### 2.5. Segmentation Metrics

The segmentation outcomes were numerically evaluated using the DSC and Jaccard index (also called Intersection over Union, IoU). The measurements are specified according to Equations (18) and (19).
(18)$$\mathrm{DSC}=2\left|S\cap T\right|\left|S\right|+\left|T\right|$$
(19)$$\mathrm{IoU}=\left|S\cap T\right|\left|S\cup T\right|$$
where *S* and *T*, correspondingly, represent the segmentation result and the ground truth [32,33]. Also, other criteria in this research, such as accuracy, sensitivity, precision, and recall, were used to evaluate the current research. The formula for calculating these indicators is presented below [34,35].
(20)$$sensitivity:\frac{TP}{TP+FN}$$
(21)$$accuracy:\frac{TP+TN}{TP+TN+FP+FN}$$
(22)$$recall:\frac{TP}{TP+FN}$$
(23)$$precision:\frac{TP}{TP+FP}$$
where *TP*, *TN*, *FP*, and *FN* are metrics at the pixel level that, in turn, stand for the confusion matrix’s true, true-negative, false-positive, and false-negative values.

## 3. Proposed SLIC-Based Deep Graph Network (SLIC-DGN)

This section explains the suggested SLIC-based Deep Graph Network (SLIC-DGN) for automatically segmenting the liver and tumors. Figure 2 depicts the development of the suggested framework.

### 3.1. Images Preprocessing

First, pre-processing is performed for each image by considering the unit Hounsfield coefficient for liver tissue, and the intensity of pixels above 150 and below zero are removed and replaced with zero. Second, resizing the 350 × 350 images is considered to focus on the liver tissue. Third, data augmentation on CT images is used to prevent the problem of overfitting during the network training phase. The images were rotated 0–20 degrees and flipped left-right and right-left for this purpose. Fourth, considering that the liver and tumor data are collected based on the same mask, for liver segmentation, a binary mode is performed between the liver and the pixels that do not contain the liver. This operation is also valid for tumor segmentation.

### 3.2. Graph Embedding Stage

The graph representation is obtained via the SLIC (Simple Linear Iterative Clustering) method. The SLIC approach clusters images and extracts a sequence of superpixels as various image regions. Figure 3 depicts how to create an image graph using the SLIC approach. Figure 3 shows that each node of this network corresponds to the superpixel regions retrieved using the SLIC approach. Furthermore, the average intensity of pixels in each region is taken as the feature vector corresponding to each area and network node. In addition, the edges of the graph are examined depending on the distance and neighborhood of each area while creating a graph adjacency matrix. Neighboring places are linked, but non-neighboring regions are not.

### 3.3. Architecture of the Proposed Method

Figure 4 illustrates the suggested method’s architecture. The following options are provided for the suggested architecture:(a)A convolutional layer on a Chebyshev graph with batch normalization, Relu activation, and batch filtering.(b)The architecture of the previous step is repeated three times.(c)A dropout layer is considered to avoid the overfitting phenomenon.(d)The flattened feature vector is fed into a dense (FC) layer.(e)Scores in the FC layer are computed using a softmax.

According to the proposed architecture, the difference in the dimensions of the layers is shown in Figure 5. The graph corresponds to the areas extracted from the SLIC method. The number of SLIC regions and characteristics in each area is used to calculate the dimensions. The developed graph used as input to the feature-encoding stage of the suggested technique features C nodes since each CT image contains C-SLIC areas. Each node in the built graph has one sample since each node’s feature vector is considered the average pixel intensity across each area. Independent of the network nodes, the input dimension of the graph convolutional layer is 16 pixels. The first graph convolution layer’s output dimension is considered to be 16; therefore, the resulting graph has C vertices and 16 samples at each node. 

The second graph convolution layer generates a C-node graph with 16 samples in each vertex; a C-node graph with the exact number of nodes is generated by the third, and so on, until the fourth graph convolution layer generates a C-node graph with a dimension of 2. The feature-encoding step’s extracted output is sent to a dropout, and the resultant C node graph is flattened with two samples in each node to create a vector of C elements, as previously discussed. After being flattened, the vector is taken via a softmax layer. Figure 5 shows that each graph convolution layer’s order of the Chebyshev polynomial expansion is unique and is determined by the coefficients K1, K2, K3, and K4 for the four specified graph convolutional layers. The suggested network’s specifications, including the size of the filters, the number of parameters, etc., are displayed in Table 1.

The tumor segmentation phases are demonstrated in Figure 6. This graphic illustrates how the trained model is fine-tuned to achieve this goal after being trained for liver segmentation. A binary mask matching the tumor area repeats all actions mentioned in Section 3.2.

### 3.4. Training and Evaluation

The suggested network in this study has been trained and evaluated using the trial-and-error approach based on 10-fold cross-validation [36]. According to this method, all samples participate in the evaluation, and the overfitting phenomenon does not occur in the training process.

Table 2 shows the optimal parameters selected to design the proposed architecture. As is clear from this table, the number of different layers, different optimization algorithms, batch size, etc., were used, and the best value was considered for the proposed architecture.

## 4. Results

The findings acquired using the suggested approach are provided in this section. The simulation results are also contrasted with those of earlier research in this section. All of the outcomes were obtained using Python programming, the Google Colab environment, and a Tesla GPU and 25 GB of RAM.

Figure 7 shows the accuracy and Dice loss of the proposed method for liver segmentation. As can be seen, 60 iterations have been considered for network training and evaluation, and the accuracy increased and the loss decreased by increasing the iterations of the algorithm so that the accuracy obtained for liver segmentation is 99%. Figure 8 illustrates the original CT image samples, corresponding binary liver mask, and predicted image of the trained SLIC-DGN for liver segmentation, respectively, from left to right.

As explained in the details of the proposed SLIC-DGN for tumor segmentation in Section 3.2 and Figure 6, we needed to fine-tune the pre-trained liver segmentation network to achieve this aim. Figure 9 represents Dice loss and accuracy convergence plots according to the binary mask of the tumor region in 600 iterations. The number of iterations specified to train the proposed SLIC-DGN for liver segmentation equals 200 in the training accuracy and loss plots. The remaining iterations are considered for fine-tuning the network to reach convergence and obtain the segmented region corresponding to the tumor tissue. Finally, the segmentation accuracy for the tumor tissue based on the test diagram is 98% in 30 iterations. The predicted image according to the binary mask of the tumor region can be seen in Figure 10.

The evaluation of SLIC-DGN in terms of accuracy, Dice coefficient, sensitivity, recall, precision, and mean IoU is shown in Table 3. As seen in this table, the evaluation metrics confirm the good performance of the network.

In order to validate the proposed model, we performed receiver operating characteristics (ROC) on the SLIC-DGN network. The results of this analysis are shown in Figure 11. As is shown, the graph obtained for liver segmentation is in the range of 0.8 to 1, which shows the optimal effectiveness of the proposed network.

According to Section 3.2, the first step of the graph embedding stage of the proposed SLIC is the extraction of regions according to the numbers dedicated to the simple linear iterative clustering algorithm. Figure 12 shows the clustering of CT images considering a different number of regions. For visual comparison, clustering according to 200, 600, and 1000 regions are shown separately. In contemplation of having a compromise between the SLIC regions and the accuracy of the segmentation, the number of 600 regions was selected to obtain the related graph. The details of evaluation parameters are represented in Figure 13, corresponding to these three different regions. This figure confirms the best effectiveness of 600 regions in terms of accuracy, Dice coefficient, and mean IoU.

To better determine the effectiveness of the suggested model for liver and tumor segmentation, the SLIC-DGN was compared with the Modified U-Net [37] and Shortcut CNN [38] networks. The two compared networks have been widely used in recent studies related to liver segmentation. The performance results in terms of accuracy and Dice loss are shown in Figure 14. According to Figure 14, as can be seen, the proposed test network has been able to achieve the highest accuracy and the lowest error. Also, as is shown, the proposed network’s fluctuation rate is much lower than that of the two compared networks, which is related to the customized architecture of the proposed network.

We have compared the effectiveness of our proposed method with recent studies. The obtained results are presented in Table 4. As mentioned in Section 1, automatic detection of the liver organ using machine learning methods has reached human accuracy (above 90%) in recent years. However, high diagnostic accuracy based on automatic machine learning methods for detecting liver tumors remains challenging. For this reason, as highlighted in Table 4, most of the recent research did not want to report the accuracy of their network due to the low effectiveness of their algorithm in detecting liver tumors, and they only reported the accuracy of their proposed method for liver detection. To find the way to the practical application of the present research, it is necessary to report the accuracy of the network both for the automatic detection of the liver organ and for the segmentation of liver tumors. Based on the obtained results, the accuracy of the proposed method for segmenting liver and liver tumors is 99.1% and 90%, respectively, which is higher than the accuracy of previous studies [26,27]. According to this table, as is shown, the proposed method based on the documentation presented in the previous graphs and curves has achieved an accuracy above 90% for the segmentation of liver tumors for the first time in present research.

CT images often have various noises for reasons such as patient movement, weakness in the device, etc. These noises can make it difficult to diagnose correctly in the relevant location and determine the dimensions and size of the tumor. In recent years, many methods have been provided for the automatic diagnosis of liver and liver tumors, which also have high diagnostic accuracy. However, none of these methods have been evaluated in noisy environments. To determine the more precise performance of the proposed model, we evaluated the proposed SLIC-DGN network along with the Shortcut CNN and Modified U-Net networks, which have recently been widely used in the segmentation of liver tumors in a noisy environment. For this purpose, Gaussian white noise was added to CT image samples in a wide range of different SNRs. Figure 15 shows an example of noise added to a CT image at different SNRs. Also, the obtained results are shown in Figure 16. As is clear from Figure 16, the proposed network achieved accuracy above 90% in SNR = −4 and shows good resistance in a wide range of SNRs. However, the popular Shortcut CNN and Modified U-Net networks have not been able to show promising results. For the present method to be used in the medical field to assist doctors, it is necessary that, in addition to high accuracy and low error in segmentation, it has good resistance in different noisy environments. As was proven, the proposed network had a high resistance to noise, which is directly related to the unique architecture proposed, including the number of layers, dimensions of filters, etc.

Despite the favorable effects of the proposed method, this study also has disadvantages, which will be mentioned below, like other studies. First, the proposed network has only been evaluated based on the LiTS17 database. For real-time segmentation, it is necessary to evaluate other valid databases. Second, to increase the data and prevent the phenomenon of overfitting, the classic way of increasing the data has been used, which can be used in future studies to evaluate and compare the effectiveness of Generative Adversarial Networks [39]. Considering the optimal accuracy of the proposed model, it can be used in the future to assist doctors and radiologists in identifying the dimensions and exact location of liver tumors.

## 5. Conclusions

This study presents an automatic method based on deep learning networks for automatically segmenting liver and liver tumors. The proposed deep model is organized based on simple linear iterative clustering and Chebyshev graph convolutional networks. The customized architecture in this study consists of four graph convolutions along with the FC layer and is evaluated on the LiTS17 benchmark database. The proposed model is compared with Modified U-Net and Shortcut CNN networks and is able to show promising accuracy. Also, the proposed model was compared with other previous studies and achieved the highest segmentation accuracy. The suggested method can be used to assist doctors and radiologists in automating liver and tumor segmentation.

## Figures and Tables

**Figure 1 sensors-23-07561-f001:**
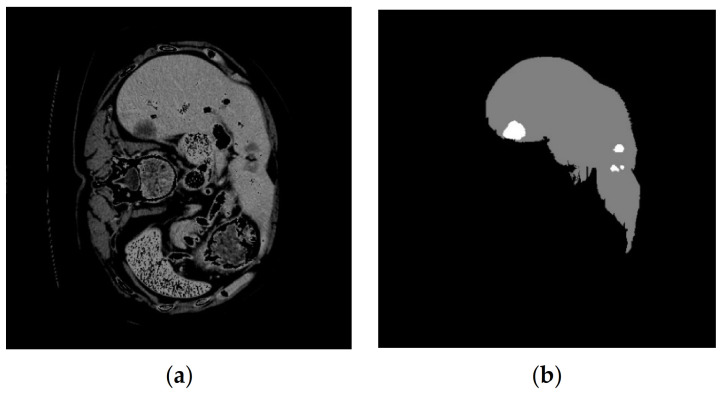
An example of a CT image from the LiTS17 database that (**a**) displays the original CT image and (**b**) shows the mask corresponding to the liver organ and the tumor [28].

**Figure 2 sensors-23-07561-f002:**
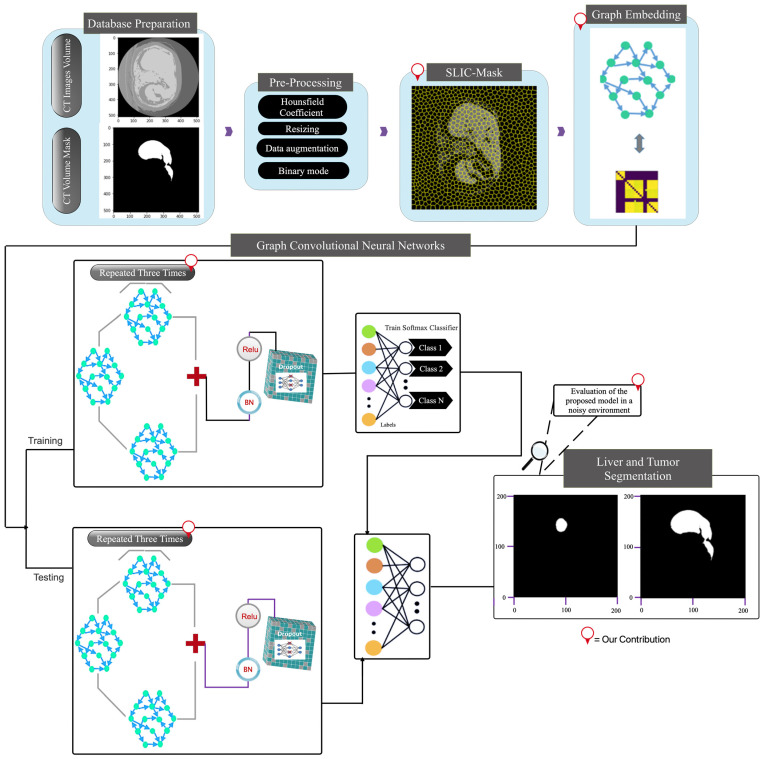
The main framework of the proposed deep model.

**Figure 3 sensors-23-07561-f003:**
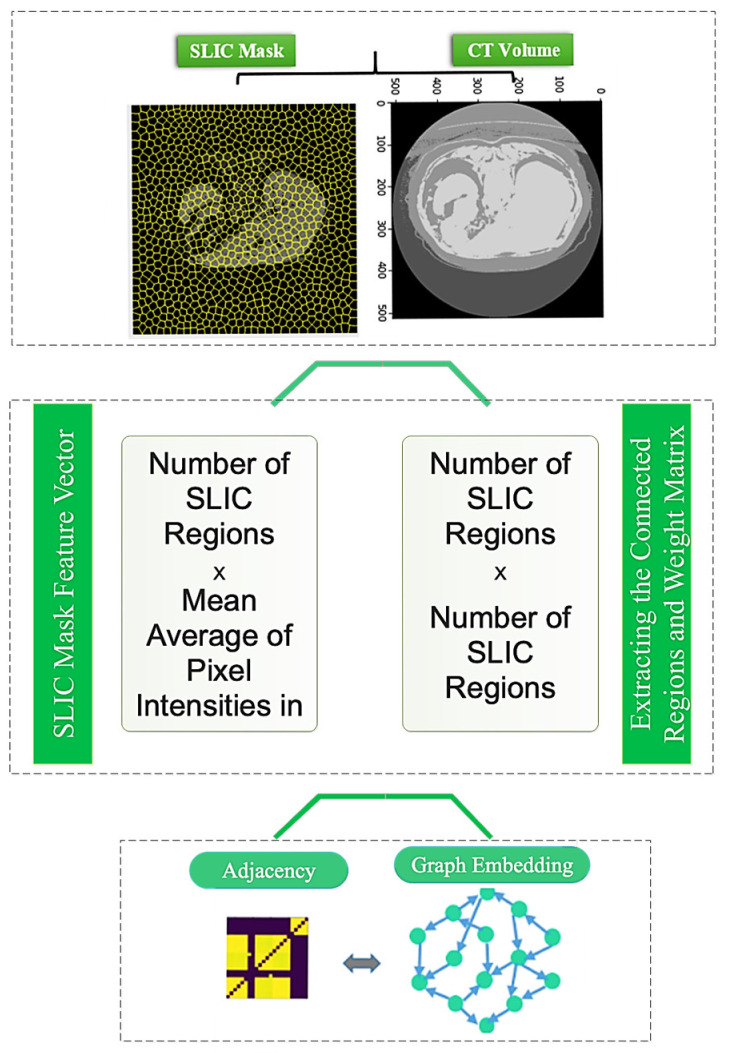
How to perform graph embedding from CT images.

**Figure 4 sensors-23-07561-f004:**
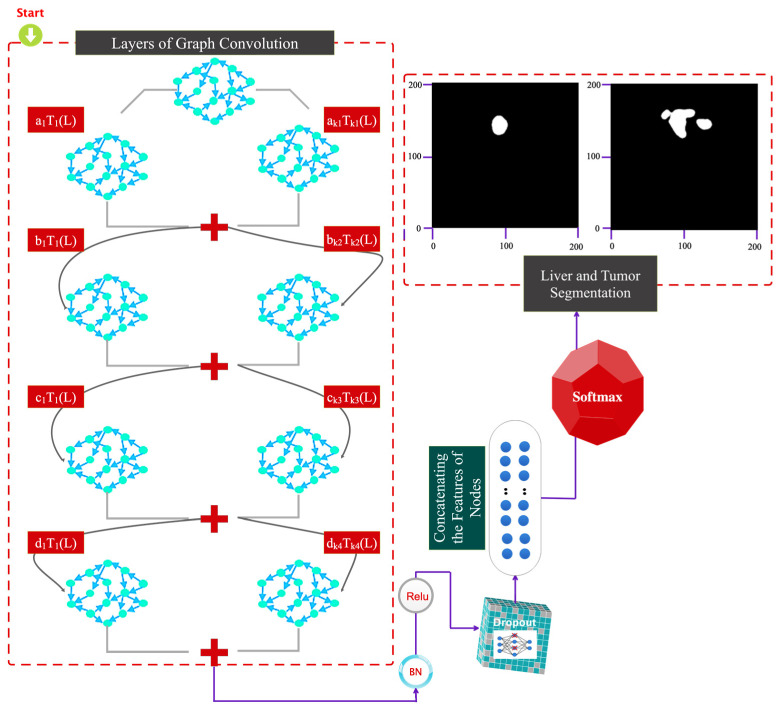
Graphic representation of the details of the proposed architectural layers.

**Figure 5 sensors-23-07561-f005:**
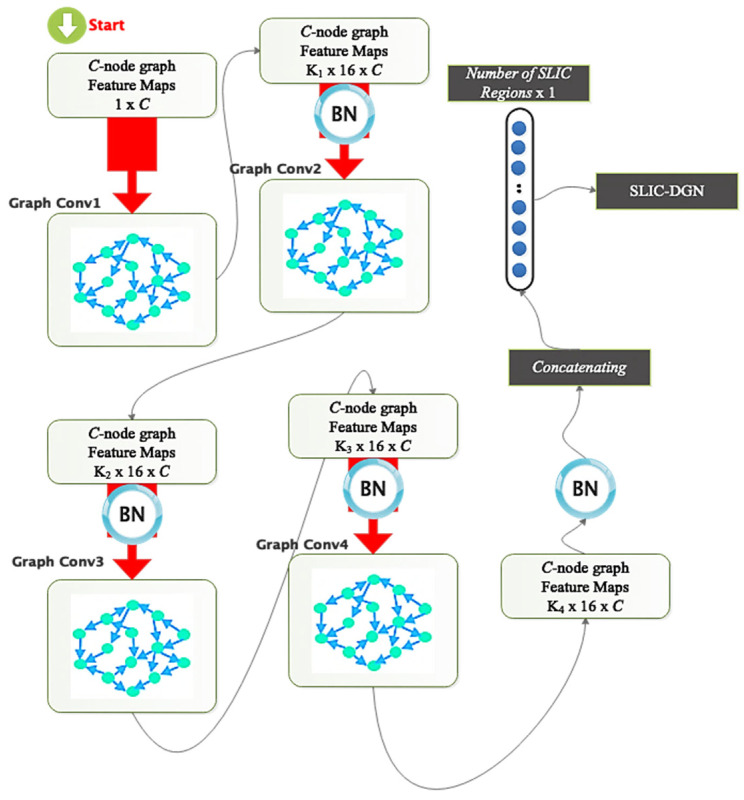
Comparison of dimensions of layers and parameters in the suggested model.

**Figure 6 sensors-23-07561-f006:**
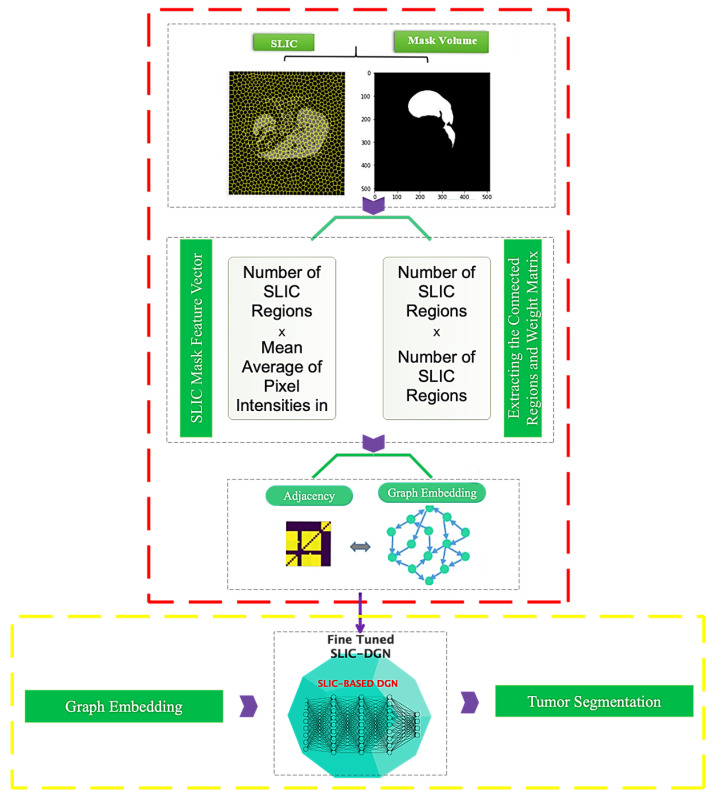
Architecture of the proposed SLIC-DGN for tumor segmentation.

**Figure 7 sensors-23-07561-f007:**
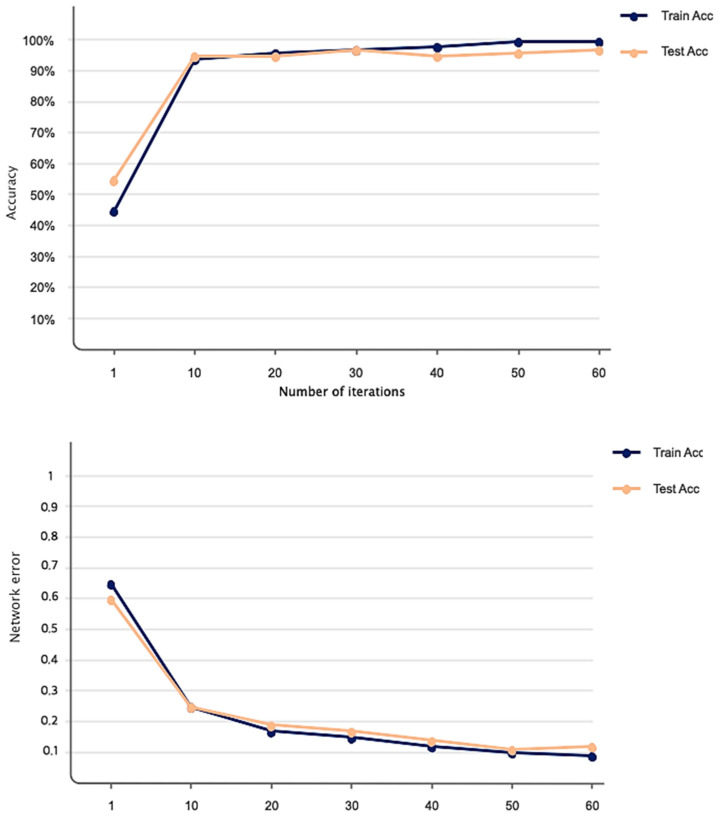
Accuracy and loss obtained based on the proposed network for liver segmentation in 60 iterations.

**Figure 8 sensors-23-07561-f008:**
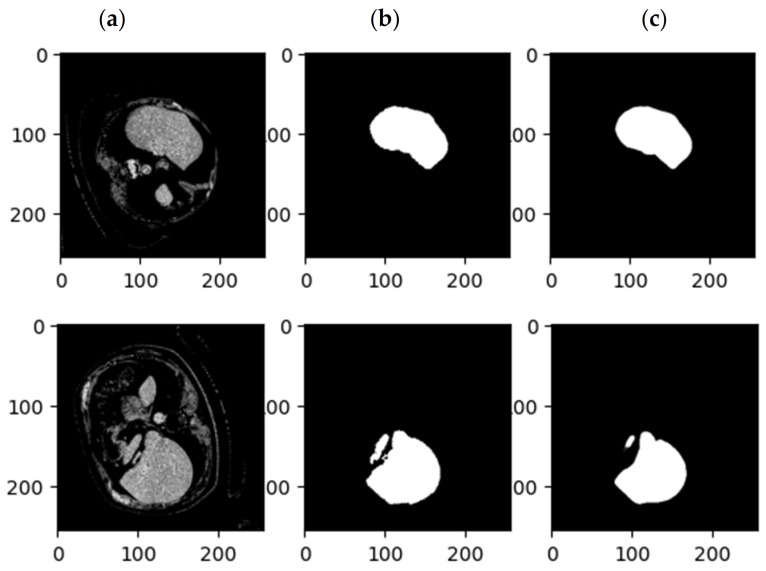
Examples of CT images related to (**a**) the original CT, (**b**) the mask related to the liver, and (**c**) the results of the proposed SLIC-DGN.

**Figure 9 sensors-23-07561-f009:**
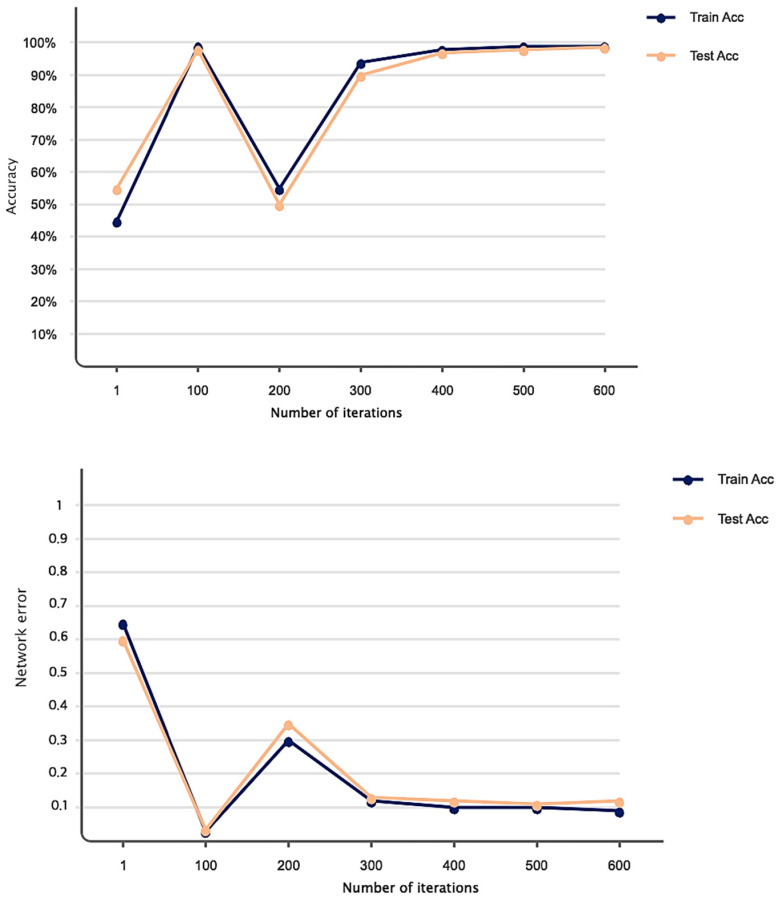
Accuracy and loss obtained based on the proposed network for liver tumor segmentation in 600 iterations.

**Figure 10 sensors-23-07561-f010:**
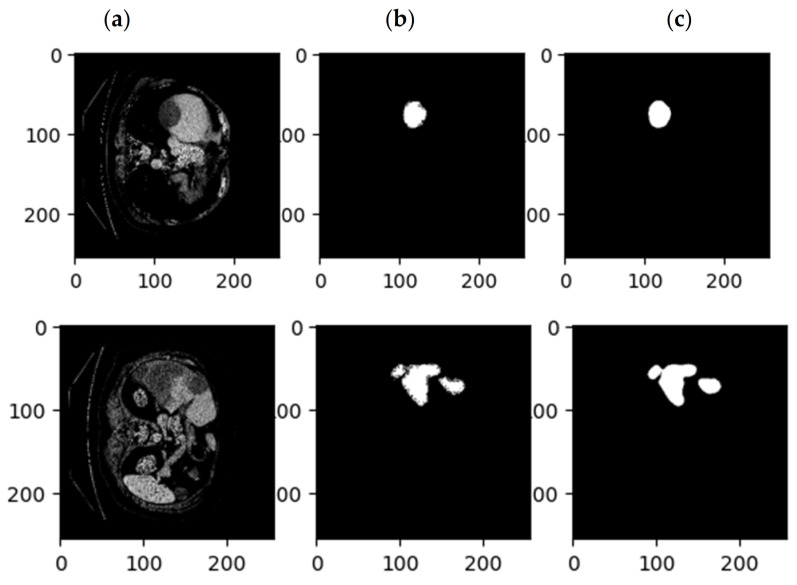
Examples of CT images related to (**a**) the original CT, (**b**) the mask related to the tumor, and (**c**) the results of the proposed SLIC-DGN.

**Figure 11 sensors-23-07561-f011:**
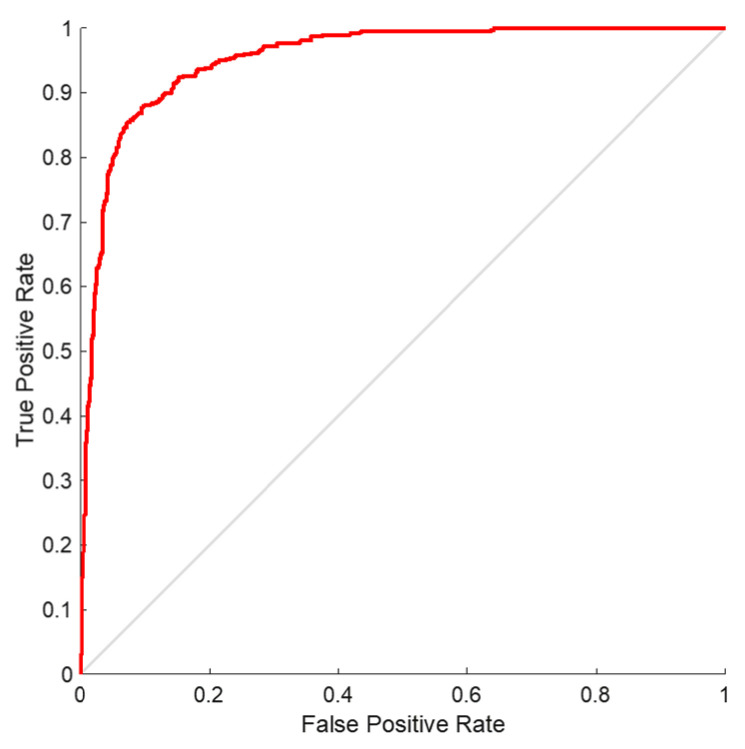
ROC curve analysis for liver organ segmentation.

**Figure 12 sensors-23-07561-f012:**
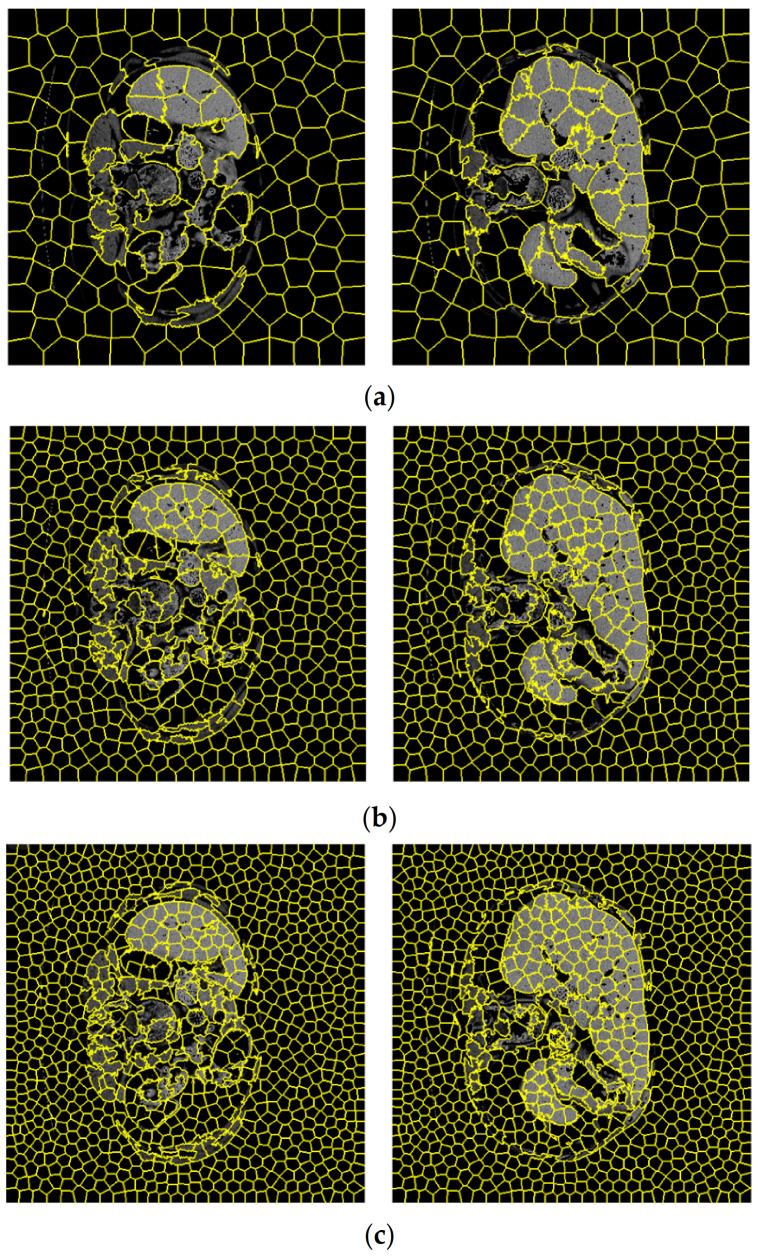
CT images corresponding to three different number of SLIC regions equal to (**a**) 200, (**b**) 600, and (**c**) 1000.

**Figure 13 sensors-23-07561-f013:**
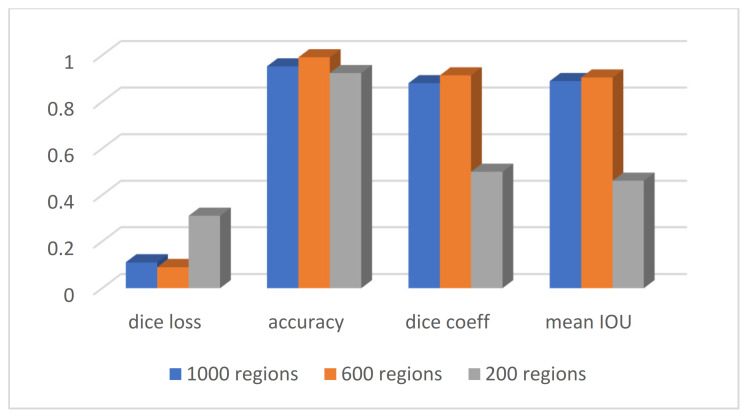
Evaluation parameters with different numbers of SLIC regions.

**Figure 14 sensors-23-07561-f014:**
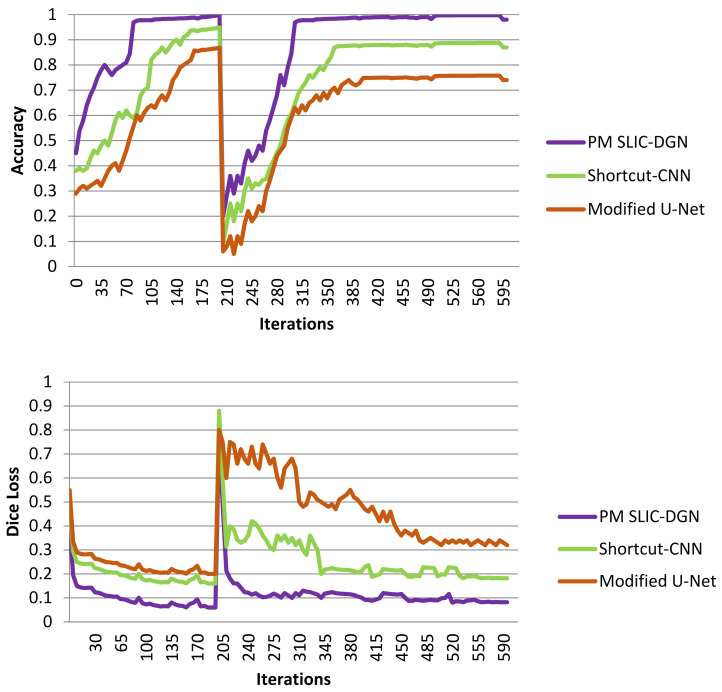
Comparing the accuracy of the proposed SLIC-DGN with modified U-Net and Shortcut CNN networks in terms of accuracy and Dice loss.

**Figure 15 sensors-23-07561-f015:**
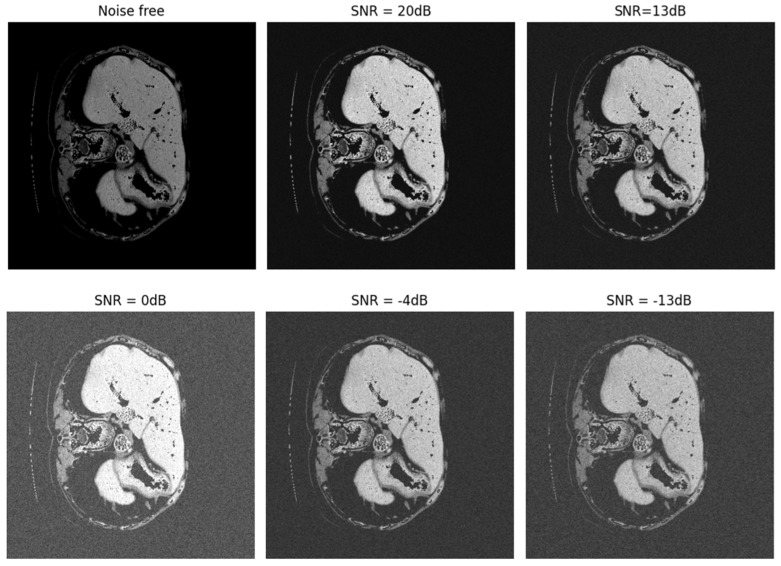
Noise added to CT images in a range of different SNRs.

**Figure 16 sensors-23-07561-f016:**
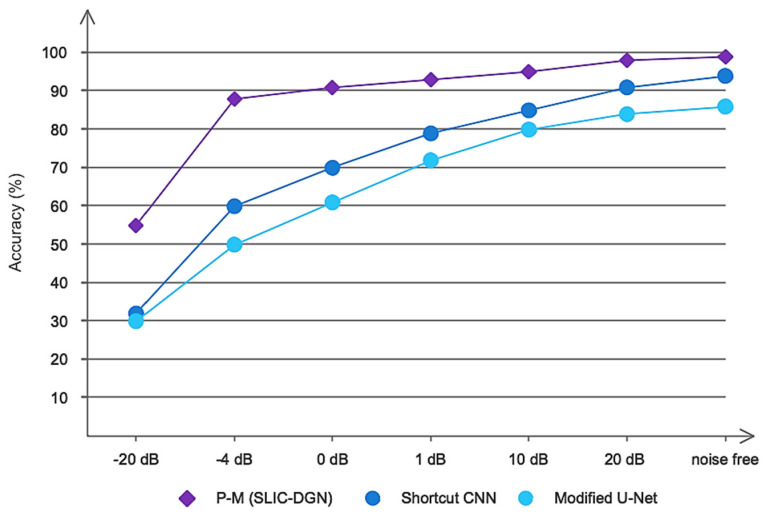
The results obtained for the proposed network and compared networks in noisy environments.

**Table 1 sensors-23-07561-t001:** Details regarding the number of layers, dimensions of layers etc., in the proposed architecture.

Layer	Shape of Weight Tensor	Shape of Bias	Number of Parameters
First graph convolution layer	[K1, 16, 16]	[16]	256 × K1 + 16
Batch normalization layer	[16]	[16]	32
Second graph convolution layer	[K2, 16, 16]	[16]	256 × K2 + 16
Batch normalization layer	[16]	[16]	32
Third graph convolution layer	[K3, 16, 16]	[16]	256 × K3 + 16
Batch normalization layer	[16]	[16]	32
Fourth graph convolution layer	[K4, 16, 2]	[2]	32 × K4 + 2
Batch normalization layer	[16]	[16]	32
Softmax layer	-	[2]	2 × C × K4

**Table 2 sensors-23-07561-t002:** Optimal settings are chosen for the suggested architecture.

Parameters	Search Space	Optimal Value
Optimizer	Adamax, SGD, Adam, RMSprop, Adadelta	Adam
Number of graph convolution layers	1, 2, 3, 4, 5, 6, 7	4
Size of ouput in first layer	8, 16, 32, 64	16
Size of output in second layer	8, 16, 32, 64	16
Siz.e of output in third layer	8, 16, 32, 64	16
Learning rate	0.1, 0.01, 0.001, 0.0001	0.001
Dropout rate	0.1, 0.2, 0.3	0.2
W.eight decay of Adam optimizer	5 × 10^−5^, 5 × 10^−4^	5 × 10^−4^

**Table 3 sensors-23-07561-t003:** Performance of the suggested SLIC-DGN for segmenting the liver and tumor.

	Liver Segmentation	Tumor Segmentation
**Accuracy (%)**	99.1	98.7
**Dice Coefficient (%)**	91.1	90
**Mean IoU (%)**	90.8	89.2
**Sensitivity (%)**	99.4	97.9
**Recall (%)**	99.4	98.9
**Precision (%)**	91.2	88.7

**Table 4 sensors-23-07561-t004:** Comparing the accuracy of the proposed method with other previous methods.

References	Method	Dataset	Liver Segmentation (%)	Tumor Segmentation (%)
Hänsch et al. [14]	3D-CNN	DCE-MRI	70	59
Dickson et al. [19]	DCMC-U-Net	LiTS17	97	80
Dong et al. [22]	Hybridized FCNN	LiTS17	97.22	87
Tang et al. [23]	E^2^Net	LiTS17, 3Diracadb01	97	74
Li et al. [24]	RDCTrans U-Net	LiTS17	93	89
Ansari et al. [25]	Res32-PAC-UNet	LiTS17	95	-
Khan et al. [26]	RMS-U-Net	3Dircadb01, LiTS17	97.50	86.70
Bogui et al. [27]	U-NeXt	LiTS17	99	-
**Proposed model**	SLIC-DGN	LiTS17	**99.1**	**90**

## Data Availability

No new data were created.

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
