# Peer review of "Automatic Liver Tumor Segmentation from CT Images Using Graph Convolutional Network"

_sensors, 2023, doi:10.3390/s23177561_

Round 1

Reviewer 1 Report

This paper presents a liver and tumor segmentation from CT images using graph convolutional network. The method is sound. However, there are some concerns about this study. This paper needs to be further refined before it can be accepted.

- The title of this paper should be refined. This title seems to imply that the liver and tumor are two independent parts in human body. It does not clearly indicate that "tumor" is the "liver tumor".

- It is unclear what is the technical challenge in the liver and tumor segmentation 

- The third paragraph is too long. It is better to reorganize it and divide it into several paragraphs.

- Lack of the analysis and discussion about the claimed contributions.

- More clinical reference should be added in the clinical background.

- More studies on medical image segmentation need to be cited to enhance the discussion of the previous works, e.g. Vessel contour detection in intracoronary images via bilateral cross-domain adaptation; Causal knowledge fusion for 3d cross-modality cardiac image segmentation. 

- It is better to indicate the claimed contributions in the flowchart (Fig.2).

- Please improve the image resolution of all figures.

- In Algorithm 1. It is hard to build the clear relationship betweem the steps in this algorithm and the contents in Section 3.

- Fig.7,8,10 occupies too much space. It is better to fuse them into one figure.

- It is not suggested to make a figure across the pages, like Fig.13.

- In Table 4, why does there lack the results of tumor segmentation for [11], [12], [15], [18], [20] ?

- Some grammatical errors.

Need moderate refinement

Author Response

Reviewer#1:

Comments:

This paper presents a liver and tumor segmentation from CT images using graph convolutional network. The method is sound. However, there are some concerns about this study. This paper needs to be further refined before it can be accepted.

  • ⎫ Thanks to the esteemed reviewer, we believe that your comments have been very useful and effective in enhancing the scientific and writing framework of the manuscript. We have considered all the comments in their entirety and made every effort to correct the manuscript in the manner suggested by the honorable reviewer.

  1. 1. The title of this paper should be refined. This title seems to imply that the liver and tumor are two independent parts in human body. It does not clearly indicate that "tumor" is the "liver tumor".
  • ⎫ The manuscript is revised based on this comment. According to the opinion of the respected reviewer, we have changed the title of the article as follows: “Automatic Liver Tumor Segmentation from CT images using Graph Convolutional Network”

Which is highlighted in Introduction section, page 1 and line 1.

  1. 2. It is unclear what is the technical challenge in the liver and tumor segmentation
  • ⎫ The manuscript is revised based on this comment. According to the opinion of the respected referee, the technical challenge for liver and liver tumors is as follows:
  • ⎫ However, detecting the liver and separating tumors from CT pictures is a difficult job that necessitates the expertise of radiologists and specialized physicians. It is al-so time demanding and mistake prone. This is due to the following reasons: First, the form and texture of the liver vary across the CT image. Second, the contrast in CT images for nearby tissues and organs is minimal. Third, different artifacts in the acquired CT images may cause the liver organ to appear vague, indistinct, and non-uniform. Furthermore, a tissue may be misdiagnosed as a tumor, or a tumor may not be identified, putting the patient's life in peril. As a result, greater care and focus are required when segmenting hepatic lesions in CT scans [4]. As a consequence, it appears essential to develop an intelligent technique for autonomously segmenting liver lesions.
  • ⎫ According to previous research, the detection accuracy for the liver organ is nearly equal to the detection accuracy for humans. However, previous research has found that the accuracy of liver tumor diagnosis is low, has limitations, and needs to be improved.

Which is highlighted in introduction section, pages 2 and 2,  lines 65-75 and 147 and 150.

  1. 3. The third paragraph is too long. It is better to reorganize it and divide it into several paragraphs.
  • ⎫ The manuscript is revised based on this comment. According to the opinion of the respected judge, the third paragraph was shortened and divided into several smaller paragraphs.

  1. 4. Lack of the analysis and discussion about the claimed contributions.
  • ⎫ The manuscript is revised based on this comment. With respect to the opinion of the respected referee, we have fully presented the limitations of the previous studies and have presented a solution to solve the relevant challenges as follows:
  • ⎫ As it is clear from the review of previous studies, despite the favorable efficiency, there are several basic limitations in these studies. The first limitation is the use of classical ma-chine learning algorithms for feature selection/extraction and segmentation. The use of these algorithms causes the selected feature vector to be not optimal for segmentation and classification. The next limitation of previous research can be considered low detection accuracy. According to previous research, the detection accuracy for the liver organ is nearly equal to the detection accuracy for humans. However, previous research has found that the accuracy of liver tumor diagnosis is low, had limitations, and needs to be improved. The next limitation of previous studies is that the algorithm used in these studies was evaluated under stable and noise-free environments, and the efficiency of these algorithms in noisy environments is not known. The current study aims to overcome the relevant challenges and overcome the limitations of previous studies. In the proposed method, the Simple Linear Iterative Clustering (SLIC) algorithm has been used to cluster CT images of the liver, and graph convolutional networks have been used to detect liver organs and segment liver tumors. According to the obtained results, the proposed method can diagnose and segment the liver and liver tumors with the precondition of high accuracy.

The contribution of this study is as follows:

• Using SLIC algorithm for optimal clustering of CT images.

• Using deep graph convolutional networks for the first time in order to segment the liver and liver tumors.

• Evaluation of the proposed network in noisy environments and maintaining the stability of the algorithm in a wide range of different SNRs.

• Providing customized architecture based on the combination of SLIC and Cheby-shev graph convolutional network.

• Achieving the highest accuracy in order to segment and diagnose the liver and liver tumors compared to previous research.

Which is highlighted in introduction, pages 3 and 4 and line 143-168.

  1. 5. More clinical reference should be added in the clinical background.
  • ⎫ The manuscript is revised based on this comment. According to the reviewer's opinion, 5 clinical references have been added to the manuscript. Accordingly, we have added clinical descriptions to the manuscript as follows:

“The liver, which aids in food digestion, is the primary organ and is located below the right ribs and behind the base of the lung [1]. It is in charge of storage, nutritional recovery, and blood cell filtration [2]. The right and left lobes of the liver are divided into two main regions. Two other lobe types are the caudate and quadrate. The etiology of hepatocellular carcinoma (HCC) is comparable to the liver cells' fast growth and potential for spreading to other parts of the body [3]. Primary hepatic cancers develop when cells behave erratically [4]. Infection frequency among men is around double that of women worldwide [5].”

Which is highlighted in introduction, page 1 and lines 36-43.

  1. 6. More studies on medical image segmentation need to be cited to enhance the discussion of the previous works, e.g. Vessel contour detection in intracoronary images via bilateral cross-domain adaptation; Causal knowledge fusion for 3d cross-modality cardiac image segmentation.
  • ⎫ The manuscript is revised based on this comment. Based on the reviewer's opinion, two mentioned studies have been added to the manuscript and discussed.

Which is highlighted in introduction, page 2 and lines 61-64.

  1. 7. It is better to indicate the claimed contributions in the flowchart (Fig.2).
  • ⎫ The manuscript is revised based on this comment. According to the opinion of the honorable judge, we have marked our contribution in Figure 2.

  1. 8. Please improve the image resolution of all figures.
  • ⎫ The manuscript is revised based on this comment. The resolution of all figures was improved based on the opinion of the respected referee.

  1. 9. In Algorithm 1. It is hard to build the clear relationship betweem the steps in this algorithm and the contents in Section 3.
  • ⎫ The manuscript is revised based on this comment. According to the opinion of the respected reviewer and another reviewer, Algorithm 1 has been removed from the manuscript due to its complexity.

  1. 10. Fig.7,8,10 occupies too much space. It is better to fuse them into one figure.
  • ⎫ The manuscript is revised based on this comment. According to the reviewer's opinion, due to the unnecessariness of Figure 7, we removed it from the manuscript. However, because Figure 8 shows the correctness and error of the liver and Figure 10 is related to the correctness and error of the liver tumor, merging the two figures can cause complexity in the understanding of the readers.

  1. 11. In Table 4, why does there lack the results of tumor segmentation for [11], [12], [15], [18], [20] ?
  • ⎫ The manuscript is revised based on this comment. According to the opinion of the respected referee, we have added the results of studies [11], [12], and, [15], to Table 4 and compared them with the proposed model. The results of studies [18] and [20] were only focused on the segmentation of the liver, and the results related to the segmentation of liver tumors were not reported.
  1. 12. Some grammatical errors.
  • ⎫ The manuscript is revised based on this comment. According to the reviewer's opinion, we rechecked the manuscript, corrected grammatical errors and highlighted.

Reviewer 2 Report

This is a well written article describing a novel method for automated  liver and tumor segmentation. It has a nice review of the literature and describes its technique well.

I only have a few comments about the style.

1.      The American Cancer Society (ACS) has recently 38 identified approximately 40,710 new cancer cases (29,200 males and 11,510 females), of 39 which 28,920 individuals (9,310 females and 19,610 males) have malignant liver and bile 40 duct cancer and will perish shortly [1]. Please clarify this better. 40,710 new cases of what? Liver Cancer? And how is this different from the 28,920? Do you mean primary vs. secondary liver cancers?

2.      The paragraph that starts with this sentence is too long and needs to be split into several smaller paragraphs.

3.      Please define what SNR’s are.

4.      SLIC

5.      DGN

6.      “As it is known, the graph obtained for liver segmentation is in the range of 0.8 to 1, which shows the optimal effective of the proposed network.” Do you mean “effectiveness”

7.      Awkward: According to Table 4, for this reason, most of the recent research did not want to report the accuracy of their network due to the low effective of their algorithm in detecting liver tumors, and they only reported the accuracy of their proposed method for liver detection. Do you mean : For this reason highlighted in Table 4, most of the recent research did not want to report the accuracy of their network due to the low effectiveness of their algorithm in detecting liver tumors, and they only reported the accuracy of their proposed method for liver detection.

8.      The suggested method can be used as an assistant to doctors and radiologists to automatic 510 liver and tumor segmentation. Awkward, do you mean “ The suggested method can be used as an assistant to doctors and radiologists to automate liver and tumor segmentation.

Author Response

Reviewer#2:

Comments:

This is a well written article describing a novel method for automated liver and tumor segmentation. It has a nice review of the literature and describes its technique well.

I only have a few comments about the style.

  • ⎫ Thanks to the esteemed reviewer, we believe that your comments have been very useful and effective in enhancing the scientific and writing framework of the manuscript. We have considered all the comments in their entirety and made every effort to correct the manuscript in the manner suggested by the honorable reviewer.

  1. 1. The American Cancer Society (ACS) has recently 38 identified approximately 40,710 new cancer cases (29,200 males and 11,510 females), of 39 which 28,920 individuals (9,310 females and 19,610 males) have malignant liver and bile 40 duct cancer and will perish shortly [1]. Please clarify this better. 40,710 new cases of what? Liver Cancer? And how is this different from the 28,920? Do you mean primary vs. secondary liver cancers?
  • ⎫ The manuscript is revised based on this comment. According to the opinion of the respected referee, it means that according to the World Health Organization's latest report in 2015, 8.8 million new cancer cases (all cancers) have been identified, of which 28,920 people have liver cancer. 

However, according to the opinion of the respected judge, a clearer sentence was proposed: “The American Cancer Society (ACS) recently identified approximately 40,710 new cancer cases (29,200 men and 11,510 women), of which 28,920 people (19,610 men and 9,310 women) have been diagnosed with malignant liver and bile duct cancer and will die soon [1].”

  1. 2. The paragraph that starts with this sentence is too long and needs to be split into several smaller paragraphs.
  • ⎫ The manuscript is revised based on this comment. According to the reviewer's opinion, the first paragraph was divided into shorter paragraphs.

  1. 3. Please define what SNR’s are.
  • ⎫ The manuscript is revised based on this comment. According to the reviewer's opinion, we have defined the abbreviations used in the manuscript in the first place used.

Which is highlighted in abstract, page 1 and lines 29.

  1. 4. SLIC
  • ⎫ The manuscript is revised based on this comment. According to the reviewer's opinion, we have defined the abbreviations used in the manuscript in the first place used.

Which is highlighted in introduction, page 4 and lines 154.

  1. 5. DGN.
  • ⎫ The manuscript is revised based on this comment. According to the reviewer's opinion, we have defined the abbreviations used in the manuscript in the first place used.

Which is highlighted in section 3, page 8 and line 275.

  1. 6. “As it is known, the graph obtained for liver segmentation is in the range of 0.8 to 1, which shows the optimal effective of the proposed network.” Do you mean “effectiveness”
  • ⎫ The manuscript is revised based on this comment. Yes, the honorable judge's opinion is completely correct, we have changed the relevant word according to the honorable judge's opinion.

Which is highlighted in page 15 and line 398.

  1. 7. Awkward: According to Table 4, for this reason, most of the recent research did not want to report the accuracy of their network due to the low effective of their algorithm in detecting liver tumors, and they only reported the accuracy of their proposed method for liver detection. Do you mean: For this reason highlighted in Table 4, most of the recent research did not want to report the accuracy of their network due to the low effectiveness of their algorithm in detecting liver tumors, and they only reported the accuracy of their proposed method for liver detection.
  • ⎫ The manuscript is revised based on this comment. Yes, the opinion of the honorable judge is absolutely correct. According to the opinion of the respected referee, the relevant sentence was modified as follows:

“For this reason highlighted in Table 4, most of the recent research did not want to report the accuracy of their network due to the low effectiveness of their algorithm in detecting liver tumors, and they only reported the accuracy of their proposed method for liver detection.”

Which is highlighted in results, page 17 and lines 430-435.

  1. 8. The suggested method can be used as an assistant to doctors and radiologists to automatic liver and tumor segmentation. Awkward, do you mean “The suggested method can be used as an assistant to doctors and radiologists to automate liver and tumor segmentation.
  • ⎫ The manuscript is revised based on this comment, Yes, the opinion of the honorable judge is absolutely correct. According to the opinion of the respected referee, the relevant sentence was modified as follows:

“The suggested method can be used as an assistant to doctors and radiologists to automate liver and tumor segmentation.”

Which is highlighted in Conclusion, page 19 and line 489.

Reviewer 3 Report

This manuscript proposes a Graph CNN based automatic liver and liver tumor segmentation method named SLIC-DGN. SLIC algorithm is applied to cluster CT liver images and deep Graph Convolutional Network is applied to segment both liver and liver tumors. The method was tested on LiTS17 database and the performance was compared with several other existing methods.

Major comments:

1. Line 162-164 motioned that “As reviewed, these networks have an accuracy of over 95% in the diagnosis of liver organs, however, they are not suitable for the segmentation of liver tumors and have a segmentation accuracy below 80%”, however the accuracy of tumor segmentation varied among different data sets. In those references, the accuracy reached >80% on some data sets. It’s not fair to directly say U-Net based methods are not suitable for this task. Better to make the description clearer.

2. Equation 23 is wrong, should be the same with sensitivity.

3. Line 323 lacks explanation, why “b) The architecture of the previous step is repeated three times.”?

4. The references in Table 4 which used different data sets to evaluate could not be compared directly with the proposed method.

Minor comments:

1. Please keep the denotation of the data set consistent through the manuscript, e.g., “LiTS” is also called “LiTS17”, “LiTS2017” in the manuscript.

2. Table 3, “for” indicates what?

3. Please only use one set of subtitles (a) (b) (c) in Figure 9 and Figure 11.

4. Wrong caption of Figure 8, and this figure is not cited in the manuscript.

5. Please keep the font of the variables consistent, otherwise it’s really difficult to read the variables and equations, including but not limited to: xˆ, U in chapter 2.2, 2.3.

6. Some extra lines in the last rectangle of Figure 3.

7. The images in the upper right corner of Figure 4 were cut and the y axis couldn’t be read.

8. According to x and y axis, the segmentation results in Figure 2 seem to have different scales?

9. The SLIC image and the mask image in Figure 6 seem to have different sizes?

Author Response

Reviewer#3:

Comments:

This manuscript proposes a Graph CNN based automatic liver and liver tumor segmentation method named SLIC-DGN. SLIC algorithm is applied to cluster CT liver images and deep Graph Convolutional Network is applied to segment both liver and liver tumors. The method was tested on LiTS17 database and the performance was compared with several other existing methods.

  • ⎫ Thanks to the esteemed reviewer, we believe that your comments have been very useful and effective in enhancing the scientific and writing framework of the manuscript. We have considered all the comments in their entirety and made every effort to correct the manuscript in the manner suggested by the honorable reviewer.

  1. 1. Line 162-164 motioned that “As reviewed, these networks have an accuracy of over 95% in the diagnosis of liver organs, however, they are not suitable for the segmentation of liver tumors and have a segmentation accuracy below 80%”, however the accuracy of tumor segmentation varied among different data sets. In those references, the accuracy reached >80% on some data sets. It’s not fair to directly say U-Net based methods are not suitable for this task. Better to make the description clearer.
  • ⎫ The manuscript is revised based on this comment. 

Yes, the opinion of the respected referee is absolutely correct, based on this, we have modified the relevant sentence as follows: 

“The next limitation of previous research can be considered low detection accuracy. According to previous research, the detection accuracy for the liver organ is nearly equal to the detection accuracy for humans. However, previous research has found that the accuracy of liver tumor diagnosis is low, had limitations, and needs to be improved.”

Which is highlighted introduction, pages 3 and line 147.

  1. 2. Equation 23 is wrong, should be the same with sensitivity.
  • ⎫ The manuscript is revised based on this comment. Yes, the opinion of the honorable reviewer is absolutely correct. Yes, the opinion of the respected referee is completely correct, the formula related to the recall was modified.

However, based on Table 6 of [*], the sensitivity formula is correct.

[*]https://www.researchgate.net/publication/323748415_Feature_Selection_Mammogram_based_on_Breast_Cancer_Mining/figures?lo=1

Which is highlighted section 2.4, pages 7 and line 270.

  1. 3. Line 323 lacks explanation, why “b) The architecture of the previous step is repeated three times.”?
  • ⎫ According to the reviewer's opinion, we have made the figures closer to the explanations for more accurate and easy reading. According to the opinion of the respected referee, the nature of deep networks is based on the repetition of layers. We have considered the architecture of the first layer of the network as follows:

a) a convolutional layer on a Chebyshev graph with batch normalization, Relu activation. 

Then we repeated the same architecture 3 times in a series according to Figure 4.

Figure 4. Graphic representation of the details of the proposed architectural layers.

We have designed the proposed network based on trial and error. Based on this, repeating 3 times will be because in this case, the network will show the best results and convergence.

  1. 4. The references in Table 4 which used different data sets to evaluate could not be compared directly with the proposed method.
  • ⎫ The manuscript is revised based on this comment. Yes, the honorable judge's opinion is completely correct, however, except for reference [11], all the previous references have reported their network performance based on the LiTS17database.
  • ⎫ However, to avoid biases and to make the comparison results fair, we separately compared our proposed method with two popular networks (the modified U-Net [30] and shortcut CNN [31] networks) that have been recently used for liver tumor segmentation and report the results in Figures 15 and 17.
  • ⎫ The following is a method for comparing and evaluating the proposed network:

“In order to better determine the effectiveness of the suggested model for liver and tumor segmentation, the SLIC-DGN was compared with the modified U-Net [30] and shortcut CNN [31] networks. The two compared networks have been widely used in recent studies related to liver segmentation. The performance results in terms of accuracy, and Dice loss are shown in Figure 15. According to Figure 15, as can be seen, the proposed test network has been able to achieve the highest accuracy and the lowest error. Also, as it is known, the fluctuation rate of the proposed network is much lower compared to the two compared networks, which is related to the customized architecture of the proposed network.

Figure 15. Comparing the accuracy of the proposed SLIC-DGN with modified U-Net and Shortcut CNN networks in terms of accuracy and Dice loss.

  • ⎫ CT images often have various noises due to various reasons such as patient movement, weakness in the device, etc. These noises can make it difficult to diagnose correctly in the relevant location and determine the dimensions and size of the tumor. Many methods have been provided for the automatic diagnosis of liver and liver tumors in recent years, which also have high diagnostic accuracy. However, none of these studies have been evaluated in noisy environments. In order to determine the more precise performance of the proposed model, we evaluated the proposed SLIC-DGN network along with the Shortcut CNN and Modified U-Net networks, which have recently been widely used in the segmentation of liver tumors, in a noisy environment. For this purpose, Gaussian white noise was added to CT image samples in a wide range of different SNRs. Figure 16 shows an example of noise added to a CT image at different SNRs. Also, The obtained results are shown in Figure 17. As it is clear from Figure 17, the proposed network has been able to achieve accuracy above 90% in SNR=-4 and show good resistance in a wide range of different SNRs. However, the popular Shortcut CNN and Modified U-Net networks have not been able to show promising results. In order for the present study to be used in the medical field as an assistant to doctors, it is necessary that, in addition to high accuracy and low error in segmentation, it has good resistance in different noisy environments. As it was proven, the proposed network had a high resistance to noise, which is directly related to the unique architecture proposed, including the number of layers, dimensions of filters, etc.

Figure 16. Noise added to CT images in a range of different SNRs.

Figure 17. The results obtained for the proposed network and compared networks in noisy environment.”

  1. 5. Please keep the denotation of the data set consistent through the manuscript, e.g., “LiTS” is also called “LiTS17”, “LiTS2017” in the manuscript.
  • ⎫ The manuscript is revised based on this comment. Yes, the opinion of the honorable reviewer is absolutely correct, we have unified the word LiTS17 in the entire manuscript.

  1. 6. Table 3, “for” indicates what?
  • ⎫ The manuscript is revised based on this comment. The word "for" was wrongly written in the table, which was removed based on the accuracy of the opinion of the honorable judge. The word "for" was wrongly written in the table, which was removed based on the accuracy of the opinion of the honorable judge.

  1. 7. Please only use one set of subtitles (a) (b) (c) in Figure 9 and Figure 11.
  • ⎫ The manuscript is revised based on this comment. Based on the opinion of the respected reviewer, we have used only one set of subtitles (a), (b), (c) in figures 9 and 11.

  1. 8. Wrong caption of Figure 8, and this figure is not cited in the manuscript.
  • ⎫ The manuscript is revised based on this comment. Thank you for the careful opinion of the respected reviewer, this error occurred during the transfer from the draft to the journal template.
  • ⎫ We have revised the caption for Figure 8 and added the relevant comments to the manuscript as follows:

“Figure 8 shows the accuracy and Dice loss of the proposed method for liver segmentation. As can be seen, 60 iterations have been considered for network training and evaluation. As can be seen, the accuracy increased and the loss decreased by increasing the iterations of the algorithm. So that the accuracy obtained for liver segmentation is 99%.”

Which is results section, pages 1 and line 356.

  1. 9. Please keep the font of the variables consistent, otherwise it’s really difficult to read the variables and equations, including but not limited to: xˆ, U in chapter 2.2, 2.3.
  • ⎫ The manuscript is revised based on this comment. Based on the reviewer's opinion, the font of the variables in section 2.2, 2.3 and 2.4 was modified.

  1. 10. Some extra lines in the last rectangle of Figure 3.
  • ⎫ The manuscript is revised based on this comment. According to the reviewer's opinion, the extra line in Figure 3 has been removed.

  1. 11. The images in the upper right corner of Figure 4 were cut and the y axis couldn’t be read.
  • ⎫ The manuscript is revised based on this comment. According to the reviewer's opinion, the Y axis in Figure 4 was modified.

  1. 12. According to x and y axis, the segmentation results in Figure 2 seem to have different scales?
  • ⎫ The manuscript is revised based on this comment. According to the opinion of the respected referee, the scales related to Figure 2 were modified.

  1. 13. The SLIC image and the mask image in Figure 6 seem to have different sizes?
  • ⎫ The manuscript is revised based on this comment. According to the opinion of the respected referee, we have modified the size of the SLIC image with the corresponding mask in Figure 6.

Reviewer 4 Report

The manuscript aims to overcome the relevant challenges and the limitations of automatic liver and tumor segmentation from CT images using a graph convolutional network.

The proposed method according to the LiTS17 dataset produced >99% accuracy.

I find the topic interesting and being worth of investigation and the document is well structured. 

Although I propose the following comments/suggestions:

- Abstract should be better organized: problem, motivation, aim, methodology, main results, further impact of those results.

- Keywords should be in alphabetical order

- Discussion should disclose the limitations of the proposed model

Author Response

Original Article Title: “Automatic Liver and Tumor Segmentation from CT Images using Graph Convolutional Network”

To: Academic Editor

Re: Response to editor comments

Dear Respected Editor,

Please find below the response to the respective academic editors’ comments. We considered all of the comments in detail and did our best to modify the paper in the way they suggested. We believe that the comments have considerably increased the quality of the manuscript. We would be most grateful if you consider the revised manuscript entitled “Automatic Liver and Tumor Segmentation from CT Images using Graph Convolutional Network” for possible publication in the MDPI publications. 

Best regards,

Best regards,

Sebelan Danishvar

Department of Eelectronic and Computer Eng

College of Eng Design and Physical Sciences Brunel University, UK.

Tel: +44 (0)1895 265089

Fax: +44 (0) 1895 265090

Reviewer#4:

Comments:

The manuscript aims to overcome the relevant challenges and the limitations of automatic liver and tumor segmentation from CT images using a graph convolutional network.

The proposed method according to the LiTS17 dataset produced >99% accuracy.

I find the topic interesting and being worth of investigation and the document is well structured. 

Although I propose the following comments/suggestions:

  • ⎫ Thanks to the esteemed reviewer, we believe that your comments have been very useful and effective in enhancing the scientific and writing framework of the manuscript. We have considered all the comments in their entirety and made every effort to correct the manuscript in the manner suggested by the honorable reviewer.

  1. 1. Abstract should be better organized: problem, motivation, aim, methodology, main results, further impact of those results.
  • ⎫ The manuscript is revised based on this comment. According to the reviewer's opinion, the abstract is organized as requested.

Which is highlighted abstract, pages 1 and lines 11.

  1. 2. Keywords should be in alphabetical order
  • ⎫ The manuscript is revised based on this comment. According to the reviewer's opinion, the keywords are organized in alphabetical order.

Which is highlighted keyword, pages 1 and lines 34.

  1. 3. Discussion should disclose the limitations of the proposed model
  • ⎫ The manuscript is revised based on this comment. According to the opinion of the honorable referee, we have presented the limitations of the current research as follows:

“Despite the favorable effective of the proposed method, this study also has disadvantages, which will be mentioned below, like other studies. First, the proposed network has been only evaluated based on the LiTS17 database. For real-time segmentation, it is necessary to evaluate other valid databases. Second, in order to increase the data and prevent the phenomenon of overfitting, the classic way of increasing the data has been used, which can be used in future studies to evaluate and compare the effective of Generative Adversarial Networks [32]. Considering the optimal accuracy of the proposed model, it can be used in the future as an assistant to doctors and radiologists in order to identify the dimensions and exact location of liver tumors.”

Which is highlighted in page 19 and line 472.

Round 2

Reviewer 1 Report

No further question.

none

Author Response

Thanks for the careful opinion of the honorable judge
I have fixed grammatical errors in the manuscript.

Reviewer 3 Report

This manuscript proposes a Graph CNN based automatic liver and liver tumor segmentation method named SLIC-DGN. SLIC algorithm is applied to cluster CT liver images and deep Graph Convolutional Network is applied to segment both liver and liver tumors. The method was tested on LiTS17 database and the performance was compared with several other existing methods.

Minor comments:

1. Please keep the font format in Equ (7) and (10) consistent with others.

2. The sample segmentation output image of the liver in Figure 2 seems not correct.

3. Considering that the proposed method was only tested on LiTS17, better to rewrite Line 167-168 to: “achieving the highest accuracy in segmenting and diagnosing liver and liver tumors on LiTS17 dataset”

4. Line 269: two ‘Accuracy’

Author Response

Reviewer#3:

Comments:

This manuscript proposes a Graph CNN based automatic liver and liver tumor segmentation method named SLIC-DGN. SLIC algorithm is applied to cluster CT liver images and deep Graph Convolutional Network is applied to segment both liver and liver tumors. The method was tested on LiTS17 database and the performance was compared with several other existing methods.

  • ⎫ Thanks to the esteemed reviewer, we believe that your comments have been very useful and effective in enhancing the scientific and writing framework of the manuscript. We have considered all the comments in their entirety and made every effort to correct the manuscript in the manner suggested by the honorable reviewer.

  1. 1. Please keep the font format in Equ (7) and (10) consistent with others.
  • ⎫ The manuscript is revised based on this comment. According to the opinion of the honorable judge, the format of formulas 7 and 10 was unified with the format of other formulas.

Which is highlighted introduction, pages 5 and 6. 

  1. 2. The sample segmentation output image of the liver in Figure 2 seems not correct.
  • ⎫ The manuscript is revised based on this comment. Yes, the opinion of the honorable judge is absolutely correct. The shape of the liver outlet was modified in the architecture of Figure 2.

Which is highlighted section 3, pages 8 and fig 2.

  1. 3. Considering that the proposed method was only tested on LiTS17, better to rewrite Line 167-168 to: “achieving the highest accuracy in segmenting and diagnosing liver and liver tumors on LiTS17 dataset”
  • ⎫ The manuscript is revised based on this comment. Yes, the opinion of the honorable judge is absolutely correct. The corresponding sentence was modified as requested.

Which is highlighted section 1, pages 4 and line 168.

  1. 4. Line 269: two ‘Accuracy’
  • ⎫ The manuscript is revised based on this comment. The corresponding sentence was corrected.

Which is highlighted section 2.5, line 269.